

# Indoor surface classification for mobile robots

Asiye Demirtaş[1,2], Gökhan Erdemir[3] and Haluk Bayram[2]

[1] Department of Electrical and Electronics Engineering, Istanbul Sabahattin Zaim University, Istanbul, Turkiye
[2] Field Robotics Laboratory-BILTAM, Department of Electrical and Electronics Engineering, Istanbul Medeniyet University, Istanbul, Turkiye
[3] Department of Engineering Management & Technology, University of Tennessee - Chattanooga, Chattanooga, TN, United States of America

Corresponding author
Gökhan Erdemir,
gokhan-erdemir@utc.edu

## ABSTRACT

The ability to recognize the surface type is crucial for both indoor and outdoor mobile robots. Knowing the surface type can help indoor mobile robots move more safely and adjust their movement accordingly. However, recognizing surface characteristics is challenging since similar planes can appear substantially different; for instance, carpets come in various types and colors. To address this inherent uncertainty in vision-based surface classification, this study first generates a new, unique data set composed of 2,081 surface images (carpet, tiles, and wood) captured in different indoor environments. Secondly, the pre-trained state-of-the-art deep learning models, namely InceptionV3, VGG16, VGG19, ResNet50, Xception, InceptionResNetV2, and MobileNetV2, were utilized to recognize the surface type. Additionally, a lightweight MobileNetV2-modified model was proposed for surface classification. The proposed model has approximately four times fewer total parameters than the original MobileNetV2 model, reducing the size of the trained model weights from 42 MB to 11 MB. Thus, the proposed model can be used in robotic systems with limited computational capacity and embedded systems. Lastly, several optimizers, such as SGD, RMSProp, Adam, Adadelta, Adamax, Adagrad, and Nadam, are applied to distinguish the most efficient network. Experimental results demonstrate that the proposed model outperforms all other applied methods and existing approaches in the literature by achieving 99.52% accuracy and an average score of 99.66% in precision, recall, and F1-score. In addition to this, the proposed lightweight model was tested in real-time on a mobile robot in 11 scenarios consisting of various indoor environments such as offices, hallways, and homes, resulting in an accuracy of 99.25%. Finally, each model was evaluated in terms of model loading time and processing time. The proposed model requires less loading and processing time than the other models.

## INTRODUCTION

With the rapid evolution of technology, robotics has moved from laboratories to almost all aspects of industry, education, health, agriculture, and life in general (*Niloy et al., 2021*). The contributions of robotics in relevant areas are remarkable in terms of mass production, safe operation, quality of education, patient satisfaction, and so on. Due to their abilities,

mobile robots are one of the most rapidly growing technologies in the world today (*Stefek et al., 2020*). Depending on their operating environment, mobile robots can be divided into two categories: indoor mobile robots and outdoor mobile robots (*Yasuda, Martins & Cappabianco, 2020*). Outdoor mobile robots are employed in various applications, such as agriculture, military, search-and-rescue, safety, and more (*Rubio, Valero & Llopis-Albert, 2019*). Due to their high capabilities and human-friendly interfaces, indoor mobile robots have become increasingly prevalent daily. Indoor mobile robots are capable of performing many tasks in a given indoor environment, including cleaning rooms (*Muthugala et al., 2020*), providing efficient human services (*Ruan, Wu & Xu, 2021*), and serving meals (*Guan et al., 2021*). However, indoor mobile robots' dexterous and safe locomotion becomes challenging on surfaces with varying features. Indoor mobile robots require environmental data to adjust their movements intelligently and react to a changing environment. Therefore, timely recognition of surface types is crucial for mobile robots to carry out a given mission successfully (*Bai, Guo & Zheng, 2019*).

The nature of surfaces varies incredibly in ordinary indoor areas. Usually, indoor floors are flat and thus easy to traverse. However, certain areas could be slippery or bumpy, which may result in difficulties in the mobility of robots or hazardous incidents. Therefore, knowledge of the floor type can aid mobile robots in traversing (moving) and adapting their movement according to the surface to avoid undesirable incidents (*Xue et al., 2022*). For instance, if the mobile robot identifies surfaces as wood floors, it can move at high speed, as the wood floor is comparatively simple and safe for movement. Certain surfaces, like carpets, are uneven and lofty, which affects movement; thus, mobile robots should slow down their speed. Therefore, the methods for estimating the current or forthcoming floors' characteristics significantly contribute to the motion capabilities of indoor mobile robots. Surface classification, depending on various sensing modes, can be divided into two major categories: contact-based classification, which includes vibration and touch, and contactless classification, which mainly makes use of vision and sound.

The classification of surface type or terrain using outdoor robots has gained attention in recent years because of the growing utilization of unmanned ground vehicle (UGV) and unmanned aerial vehicle (UAV) in both civilian and military applications; however, as compared to outdoor robotics, the research for indoor surface classification requires more attention. Researchers in prior studies used sensors, cameras, or a hybrid system (sensor and camera both) to detect the surface type. For example, *Tick et al. (2012)* classified the indoor surface using data collected from an inertial measurement unit (IMU) connected to the robot. Unlike other sensor-based studies, they used additional attributes such as velocity and acceleration to construct a dataset of 800 features which are later reduced by exploiting sequential forward floating feature selector. Then, they used linear Bayes to classify surfaces into five types (carpet, terrazzo, linoleum, ceramic tiles-A, ceramic tiles-B) and obtained an accuracy of 90% for a robot moving for 20 min.

Similar to previous work, *Bermudez et al. (2012)* utilized vibration data obtained by an IMU installed on a legged robot. The authors also incorporated magnetic encoders and back-EMF sensors to generate motor control data for better surface classification. They used 75% of collected data to train support vector machine (SVM) and attained an overall

accuracy of 93.8% in test phase to distinguish three types of surfaces (carpet, tile, and gravel). *Kertész (2016)* performed several sensing modalities (ground contact force sensor, motor force sensor, infrared sensor, accelerometer, *etc.*) on a Sony ERS-7 mobile robot to identify six types of indoor terrains with the help of a random forest (RF) classifier. Beside this, fast Fourier transformation scheme was exploited on obtained data and later system was tested. The study achieved an overall accuracy of 94% while moving the robot with barefoot and wearing socks.

On other hand, *Giguere & Dudek (2011)* used a low-speed wheeled robot with an accelerometer to train an artificial neural network (ANN) classifier to classify ten types of outdoor and indoor surfaces. They analyzed acceleration patterns with eight extracted features in the time and frequency domain to obtain an accuracy of 94.6% and 89.9% for windows of 4 s and 1 s, respectively. Unlike others, they distinguish between tiled and untiled linoleum surfaces. Instead of using motion or vibration sensors, *Bosworth et al. (2016)* utilized touch sensors to measure surface friction and impedance to enable the MIT Super Mini Cheetah robot to move over variable terrains with better locomotion. Previous studies used wheeled or legged robots with different combinations of sensing modalities to classify indoor surfaces based on vibration. However, even an increasing number of sensors does not significantly contribute to achieving higher performance. Moreover, most of these works employed traditional machine learning algorithms, whereas vision-based modalities with advanced deep learning-based models could be more effective for surface classification. In addition to vibration and motion sensing modalities, researchers have also leveraged vision-based data or hybrid datasets (vibration and images) to classify surfaces and terrains.

For instance, *Weiss, Tamimi & Zell (2008)* utilized imaging data and vibration measurements to classify 14 types of surfaces, including mix grass-gravel, mowed grass, medium grass, short grass, small bushes, dirt, clay, circular, paving, quadratic paving, tiles, coarse gravel, fine gravel, asphalts, and indoor surfaces. They first trained an SVM model on the given datasets individually and later widened the research by combining vibration data with imaging data for better classification. It is noted from the conclusion that the SVM model trained on the fused dataset outperformed the models trained on individual datasets, achieving an overall accuracy of 87.04%. Even though they used the camera to capture the images, the paper focuses only on outdoor surface classification as they grouped all types of indoor surfaces into one label ('indoor'). In contrast, we aim to classify indoor surfaces into three different labels.

Similarly, *Kurobe et al. (2021)* used audio features beside images to identify indoor and outdoor surfaces. They clustered surface types by training the convolutional neural network (CNN)-based ResNet50 model using audio-visual modality with the help of a microphone and camera installed beneath the mobile platform. Their developed scheme achieved an overall accuracy of 80%. Nonetheless, including audio modality does not enable the model to outperform vibration-based classification models. Therefore, there is a need to suggest an accurate indoor surface classification system with minimal equipment and computational cost. *Guan et al. (2022)* proposed a framework that utilizes visual and inertial perception by using camera and IMU sensors to classify multiple outdoor and some indoor surfaces, such

as carpet and tile, for mobile robot navigation. The presented framework uses CNN-based EfficientNet-B0 model to extract the image features. The obtained results demonstrate the efficacy of the proposed model for surface classification tasks. However, the indoor surface types in our study are more diverse, including a wide range of carpet, tile, and wood types in various colors and patterns. Additionally, our dataset consists of samples collected from different indoor environments, unlike previous studies which focused on a single site.

Figure 1 illustrates our approach to solving the problem, where the collected dataset is pre-processed and divided into train, test, and validation sets. We trained the dataset with eight models, including seven state-of-the-art models and our proposed modified model. The best model weights were determined by training each model with seven different optimizers.

Previous studies on surface classification have certain limitations. Firstly, the generalization of the trained classification networks is not specified, whereas models were trained over finite samples that may cause over-fitting or underfitting. Secondly, most works did not present the time analysis to detect the sudden changes in the surface. Thirdly, they lack a discussion of model efficiency regarding training time. Fourthly, the presented results include limited performance evaluation metrics. Fifthly, most studies exploited specific machine learning algorithms. Still, they lack a comparative analysis of the proposed model with state-of-the-art models, which may perform better than the suggested ones. This study aims to improve surface recognition to stabilize the mobile robot's movements so that it performs the given task more comprehensively. Besides exploiting pre-trained state-of-the-art deep learning models, the study proposed a modified MobileNetV2 to classify the surface efficiently. The key contributions include:

- Generation of a unique dataset that is composed of surface images captured in different environmental conditions.
- Unlike prior studies, our new dataset includes carpet surfaces besides wood floors and tiles.
- Comparison and analysis of the pre-trained deep-learning models for a surface classification task on the new dataset.
- Proposing a modified MobileNetV2 to establish its impact on the classification performance of robotic systems with limited computational capacity.
- Comparison and analysis by examining the effect of several optimizers (SGD, RMSProp, Adam, Adadelta, Adamax, Adagrad, Nadam) to distinguish the most efficient network for the problem of surface recognition.
- Providing a systematic evaluation of each utilized model using various optimizers based on accuracy, precision, recall, and F1-score.

The article is further organized as follows: In the next section, we explore detailed information about the generated dataset and deep learning techniques. In the comparison section, we present a comprehensive comparison of the models used. In the last section, we conclude the article with future works.

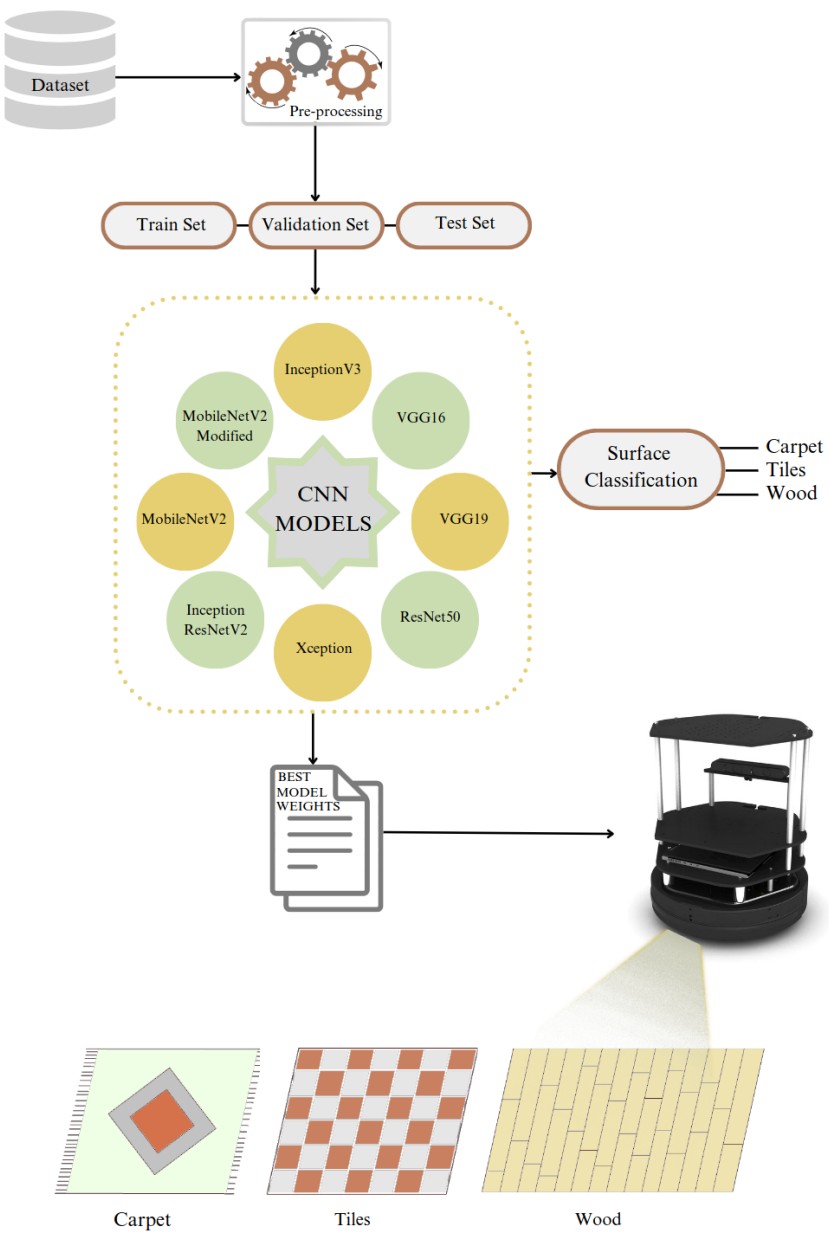

**Figure 1 Problem illustration.**

## CNN BASED SURFACE CLASSIFICATION

This section presents a detailed description of the dataset generated in this study, along with an in-depth discussion about the selection and architecture design of pre-trained state-of-the-art feature-extraction models, as well as the application of various optimization algorithms. It specifically focuses on the generation of a new dataset, pre-processing steps, implementation of CNN-based models, and the selection of the appropriate optimization algorithms. The overview of the proposed approach is illustrated in Fig. 2.

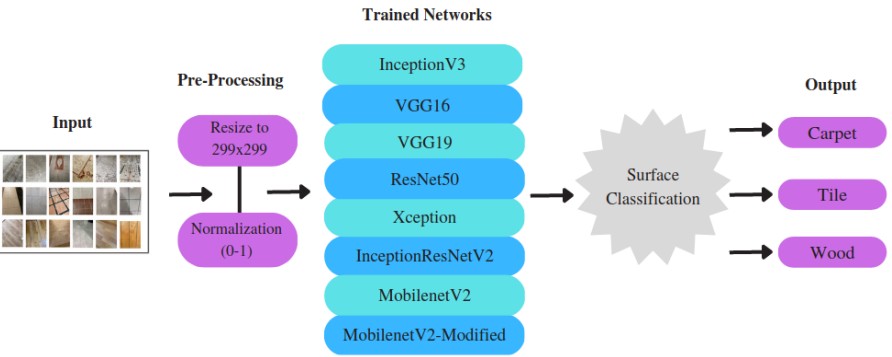

**Figure 2** **The overview of the surface classification structure.**

## Pre-processing

Carrying out a few pre-processing steps before feeding the data into DL models plays a critical role in improving performance and feasibility. To meet the fixed-size input requirement of applied and proposed CNN models, we resized the input size of the samples to 299 × 299 pixels.[1] We then performed min-max normalization, which reduced the values of a given set to a ratio between 0 and 1. Min-max normalization is expressed by Eq. (1) (*Rajendran et al., 2020*):

$$I_{out} = (I_{in} - Min)\frac{newMax - newMin}{Max - Min} + newMin \tag{1}$$

where $I_{i_n}$ denotes the original surface image, *Min*, and *Max* demonstrates the minimum and maximum intensity values, which are in the range between 0 and 255; after the min-max normalization, the resulting image is denoted by $I_{out}$, and the new minimum and maximum values are represented by *newMin* and *newMax*, respectively.

## Dataset

Datasets play a significant role in deep learning (DL)-based techniques, and accurate and precise results cannot be achieved without consistent and meaningful data. As there is a lack of an open-source indoor surface image dataset, we generated a new dataset that includes three distinct floor types: carpet, tiles, and wood. We captured dataset samples with cameras in various indoor environments and lighting conditions. Before building the dataset, we analyzed the overall dimensions of the indoor robots. Thus, while generating the dataset, we took images from different angles to ensure convenience for the indoor mobile robot's camera position. According to our literature review, the generated dataset is one of the most comprehensive datasets used in such studies. In particular, to prevent the lack of datasets in which the diversity of indoor surfaces is limited, samples in our dataset were captured in more than 20 indoor environments, including various carpet, tile, and wood floors. The samples from the generated dataset are shown in Fig. 3. The dataset can be downloaded from *Demirtas, Erdemir & Bayram (2023)*.

The generated dataset consists of a total of 2,081 samples, of which 870 samples are carpet, 638 tiles, and 573 wood floors. All the images are RGB, and the size of the collected

[1]The reason behind why we chose 299 × 299 resolution for images in our dataset is that the Inception model we used in our paper requires specifically this resolution (*Google Cloud TPU, 2023*). Furthermore, based on the Keras documentation (*Sayak, 2023*), it is a typical practice to downsize images to smaller dimensions (such as 224 × 224 or 299 × 299, *etc.*) both to facilitate mini-batch learning and to accommodate computational constraints.

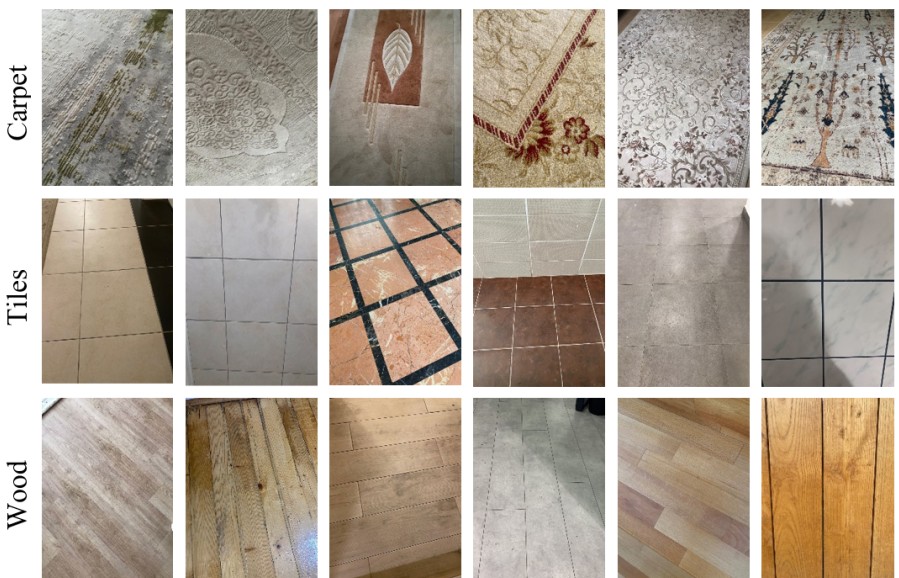

**Figure 3  Samples from the different surface types.**

**Table 1  Details of the generated dataset.**

| Dataset | Carpet | Tiles | Wood | Total |
|---|---|---|---|---|
| **Train** | 698 | 510 | 457 | 1,665 |
| **Test** | 86 | 64 | 58 | 208 |
| **Validation** | 86 | 64 | 58 | 208 |
| **Total** | 870 | 638 | 573 | 2,081 |

images varies, but before feeding to the CNN models, samples are resized to an equivalent dimension. Note that 80% of the dataset was used for training, 10% for validation, and the remaining 10% for testing. Detailed information about the generated dataset is presented in Table 1.

## Surface classification using CNN-based models

Convolutional neural networks (CNNs) are a deep learning (DL) based approach that is extensively adopted for image recognition and classification tasks in various areas (*Huang et al., 2017*; *Szegedy et al., 2015*). Instead of manually extracting certain features, as is the case with traditional methods, CNNs can automatically learn specific features from original images without the need for human supervision. Generally, a CNN architecture consists of three main layers: the convolutional layer, the pooling layer, and the dense layer. The descriptions of each layer are given in the following.

### *Convolutional layers*

The most crucial part of CNN is the convolutional layer, which uses various convolution kernel sizes to extract the particular features of a given image. A set of feature maps can be obtained by applying a convolutional layer in a given image a few times. Assuming that $F_i$

represents the CNN feature map for the $i$th layer, $F_i$ is given as:

$$F_i = \phi(F_{i-1} W_i + b_i) \tag{2}$$

where $F_i$ indicates the current network layer's feature map, $F_{i-1}$ represents the previous layer's convolution feature, $W_i$ represents $i$th layer's weight, $b_i$ is the offset vector of the $i$th layer, and $\phi$ represents the rectified linear unit (ReLU) activation function.

### Pooling layers

After applying a convolutional layer, pooling layers reduce the feature dimensionality, thus drastically minimizing the computational complexity. In addition, pooling layers effectively control the risk of over-fitting. The output feature of the $h$th local receptive field in the $l$th pooling layer can be calculated as follows:

$$x_h^l = downsample(x_h^{l-1}, s) \tag{3}$$

where *downsample* indicates the down-sampling function, $x_h^{l-1}$ represents the feature vector of the previous layer, and $s$ denotes the pooling size.

### Dense layers

Finally, a fully connected layer to perform the classification task collects all the features extracted by previous layers. The Softmax function generally performs class prediction with all the gathered features. The Softmax function can be expressed mathematically as:

$$softmax(z)_j = e^{z_j} / \sum_{k=1}^{K} e^{z_k}, \forall j \in \{1, \dots, K\} \tag{4}$$

where $K$ denotes the dimension of vector $z$.

## CNN-based models

In this study, several CNN-based models were used to perform the surface classification tasks, including InceptionV3, VGG16, VGG19, ResNet50, Xception, InceptionResNetV2, MobileNetV2, and a modified version of MobileNetV2. The description of each model is presented in detail below.

### InceptionV3

The InceptionV3 is a CNN based DL model that is used for image recognition tasks (*Szegedy et al., 2016*). It is an improved version of the main Inception V1 model. In the InceptionV3 model developed by Google, a group normalization layer and a fully connected layer have been added to make the network more efficient. This architecture has shown promising results on the ImageNet dataset and is widely used for image classification. The improved version aims to perform well even on applications with limited computational costs. For instance, a $3 \times 3$ convolution is divided into $3 \times 1$ and $1 \times 3$ convolutions, which allows for more effective extraction of specific details present in the image and reduces the numerical parameters of the model, resulting in shorter training times. The total depth of the model is 47 layers. Despite its more profound architecture than previous versions, InceptionV3, with the new factorization ideas, reduces the parameters without decreasing the network performance.

### VGG16

VGG16 as a CNN-based model, was ranked among the top 5 models in the ImageNet Large Scale Visual Recognition Challenge (ILSVRC) (*Russakovsky et al., 2015*) competition on ImageNet dataset in 2014 with a 7.3% error rate (*Simonyan & Zisserman, 2014*). The architecture comprises 16 layers with approximately 130 million parameters, applying $3 \times 3$ filters. VGG16 focuses on $3 \times 3$ filtered convolution layers with a stride of 1 and a $2 \times 2$ filtered max-pool layer instead of many hyper-parameters. The architecture's fundamental components are convolutional layers with varying depths, followed by three fully connected layers. The first and second dense layers contain 4,096 neurons, while the third has 1,000 neurons. Finally, the last layer is the softmax layer, where classification is performed.

### VGG19

The overall concept of the VGG19 architecture is the same as VGG16, except for the adoption of three additional convolutional layers. In this case, the first 16 convolutional layers extract features, while the following three fully connected layers are used for classification. The feature extraction layers are divided into five groups, each followed by a max pooling layer. The input image size of the VGG19 model is the same as VGG16, which is $224 \times 224$ pixels.

### ResNet50

The classification accuracy of deep CNN models increases correspondingly with the number of network layers. However, as the network depth increases, the model's training and test error rate also increases. This phenomenon is referred to as the "vanishing gradient". To overcome this problem, the Residual Network was introduced (*He et al., 2016*). ResNet50 deploys a technique called skip connections, allowing the network to link directly to the output by skipping several training layers. The main architecture of ResNet50 is inspired by the VGG architecture, consisting of five phases, each with an identity block with three convolutional layers and a convolutional block with three convolutional layers. A downsampling convolutional layer with a stride of 2 performs down-sampling. The network finalizes with fully connected layers.

### Xception

As an inspired version of the Inception model, Xception architecture was introduced by *Chollet (2017)* in 2017. The architecture of the Xception model consists of deep and comprehensive convolutional layers which operate in a collateral manner. Thus, the feature extraction process of the network is realized with a total of 71 convolutional layers. In Xception, the convolutional layers are arranged into modules and encircled by linear residual connections, making it a stack of depthwise separate convolutional layers covered by residual connections. This makes the architecture very simple to describe and perform, unlike Inception V2 or V3, which are substantially more difficult to specify. The pre-trained model was trained on the ImageNet dataset on millions of images, offering high efficiency in image recognition.

### InceptionResNetV2

The InceptionResNetV2 module is a combination of Inception and residual networks, showing promising results in image classification and object detection tasks (*Szegedy et al., 2017*). From a particular view, each InceptionResNet module can be demonstrated as a tiny CNN. These small CNNs, together with additional network layers like pooling and convolutions, compose the main architecture of the InceptionResNet network. The architecture of the InceptionResNet module consists of two main parts: shortcut connections and residual blocks. Input features are directly mapped to output features by the shortcut connection, so residual blocks estimate the residual function. Finally, the module output is composed of the corresponding map of input features as well as the output of the residual block.

### MobileNetV2

MobileNetV2 model is utilized in many computer vision applications, but mainly it offers high robustness performance in object recognition and segmentation tasks. It increases the state-of-the-art performance of mobile models and computationally limited devices across various benchmarks, activities, and model sizes (*Sandler et al., 2018*). On the main architecture of MobileNetV2, bottleneck levels, inverse residual structures, and connections are presented. This methodology makes real-time classification possible on computationally limited devices or smartphones. More detailed information about the fundamental architecture of MobileNetV2 can be found in *Sandler et al. (2018)*.

### MobileNetV2-Modified

The fundamental MobileNetV2 architecture inspires our proposed modified MobileNetV2 model. Once the core convolutional architecture has extracted the input images' features, the global average pooling layer was used to minimize the size of the extracted features. So, the classification task is done in the fully connected layer. The original MobileNetV2 model has a lower computational cost than other CNN-based models. Thus, this makes the MobileNetV2 model more effective in mobile robot-based applications. In this study, since the surface classification performance of the mobile robot was analyzed, it is more consistent to modify the MobileNetV2 model compared to other models in terms of computational cost. While the original MobileNetV2 model has an average weight of around 40 MB, which is relatively small, its architecture is changed to reduce the mobile robot's computational cost and make it work more efficiently. To achieve this, we replace the last 11 layers before the fully connected layers of the original MobileNetV2 with a 2D-convolutional layer with 16 filters with a kernel size of 5x5, ReLu as activation, and the same padding. Thus, the number of blocks was reduced from 17 to 16, as shown in Figs. 4 and 5. The convolutional layer is followed by a Global Average Pooling layer and a Dropout layer with a value set to 0.2. Then, two dense layers are added, having a size of 2,096 and 3, respectively. Softmax is used for activation to classify the three types of surfaces correctly. The size of the extracted features is minimized using the Global Average Pooling layer, and overfitting was prevented by adding a new Dropout layer to the modified architecture. Apart from removing blocks, adding dropouts, building a convolution layer, we made a significant change to the proposed model by making the last 23 layers trainable (Fig. 5),
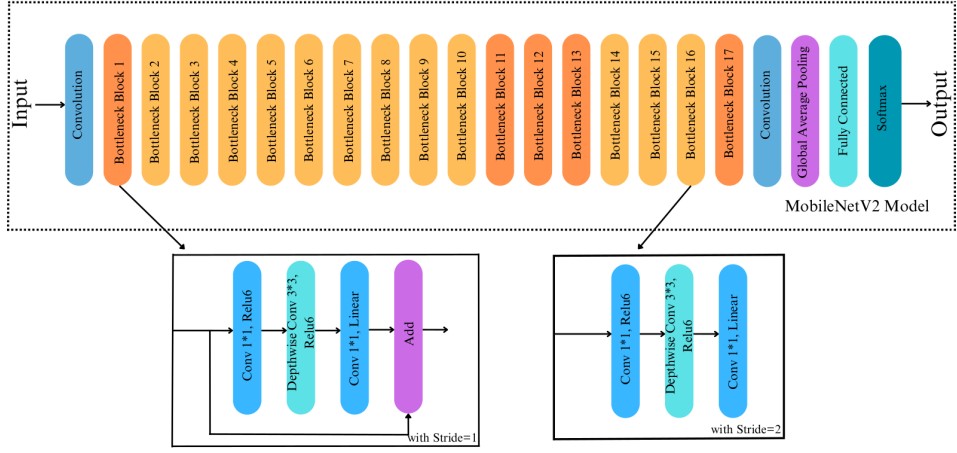

**Figure 4** Original architecture of MobileNetV2.

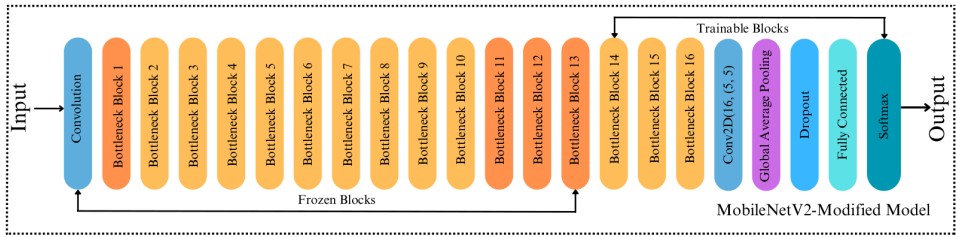

**Figure 5** Architecture of MobileNetV2-Modified.

whereas in the original MobileNetV2, all layers are frozen except for the fully connected and global average pooling layers. In the MobileNetV2 model, the frozen layers are trained on the ImageNet dataset, while the trainable layers are trained with a custom dataset. By doing so, we made 23 layers of the modified model trainable. This indicates that if we did not make any modifications to the architecture, the number of trainable parameters in the modified model would be higher than that of the original model. However, we removed the last 11 layers of the model and downsized the filter size of the last convolutional layer in the architecture. Therefore, we simplified the model, resulting in fewer trainable parameters, less training time and smaller model weight size. Note that the modification of the architecture was finalized after several rounds of fine-tunings.

## Optimizers used in training

Optimizers regulate the learning speed and the neural network weights to provide the best model performance and reduce the losses experienced during training. In other words, the optimization algorithms minimize the loss during training and ensure accurate dataset training. The optimizer algorithms applied in this study are briefly explained below.

### Stochastic Gradient Descent

Stochastic gradient descent (SGD) (*Robbins & Monro, 1951*) and its variants are frequently used in deep learning. SGD is different from the classic vanilla gradient descent technique in that it computes the gradient of the objective function. It updates the parameters on subsets of the training data instead of the complete training set. The main goal of SGD is to minimize computational costs. It is worth noting that gradient descent is a computationally intensive process by itself, especially when there is a large training set. The parameter update of SGD is done as follows (*Zhao et al., 2019*):

$$\theta_t = \theta_{t-1} - \eta \cdot \nabla_\theta J(\theta_{t-1}) \tag{5}$$

where $\theta_t \in R^d$ is the parameter vector during iteration $t$, the learning rate is represented by $\eta$, $J(\theta_{t-1})$ is the loss function defined by the model's parameters $\theta_t$ at the $t$-th iteration, and $\nabla_\theta J(\theta_{t-1}) = \partial J(\theta_{t-1})/\partial \theta$ represents the gradient of the loss function to parameters at the $(t-1)$th iteration.

### Adaptive Moment Estimation

The advantages of Adagrad and RMSprop optimizers are combined in adaptive movement estimation (Adam), which calculates the learning rates adaptively for various variables. It has a low memory demand and a high computational efficiency and calculates the learning rates for each parameter. Momentum is directly integrated with Adam to estimate the gradient's first-order moment (*Soydaner, 2020*). To account for the initialization at the origin, Adam applies bias corrections to the estimations of the first-order and second-order moments (*Kingma & Ba, 2014*). To estimate the moments, Adam employs the exponential moving average, which is computed on the gradient evaluation with respect to the current mini-batch. The exponential decay rates of these moving averages are controlled by the hyperparameters $\beta_1$ and $\beta_2$, which are usually set close to 1. $\beta_1$ and $\beta_2$ are typically set to values close to 1. The estimate of the gradient's mean, $m_t$, is calculated as:

$$m_t = \beta_1 m_{t-1} + (1-\beta_1)g_t \tag{6}$$

where $g_t$ is the gradient on the current mini-batch at step $t$. Subsequently, the estimate of the gradient's uncentered variance, $v_t$, is calculated as:

$$v_t = \beta_2 v_{t-1} + (1-\beta_2)g_t^2 \tag{7}$$

In the first stage, $m_t$ and $v_t$ are smoothly biased to starting value. Therefore, bias correction must be calculated for both the first and second moments, so the following steps are performed:

$$\hat{m}_t = \frac{m_t}{1-\beta_1^t} \tag{8}$$

$$\hat{v}_t = \frac{v_t}{1-\beta_2^t} \tag{9}$$

Finally, the parameters are updated as follows:

$$\theta_t = \theta_{t-1} - \frac{\eta}{\sqrt{\hat{v}_{t-1}}+\epsilon}\hat{m}_{t-1} \tag{10}$$

where $\eta$ is the learning rate and $\epsilon$ is a small value to avoid division by zero.

### Root Mean Square Propagation

The root mean square propagation (RMSProp) algorithm was proposed by Geoffrey Hinton (*Hinton, Srivastava & Swersky, 2014*). RMSProp optimizer is similar to the gradient descent algorithm with momentum. By employing a moving average of the squared gradient to normalize the gradient by considering the size of recent gradient descents, RMSProp attempts to address the drastically decreasing the learning rates of Adagrad. Therefore, the algorithm progresses horizontally, with larger steps converging faster as the learning rate increases (*Mukkamala & Hein, 2017*). The moving average of the squared gradient is maintained as follows:

$$\theta_t = \theta_{t-1} - \frac{\eta}{\sqrt{E[g^2]_{t-1} + \epsilon}} g_{t-1} \tag{11}$$

### Adagrad

Adagrad is an optimization technique that modifies the learning rates of the model parameters on an individual basis (*Duchi, Hazan & Singer, 2011*). The learning rate of the parameters with the biggest partial derivative of the loss decreases quickly, whereas the learning rate of the parameters with smaller partial derivatives decreases more slowly (*Goodfellow, Bengio & Courville, 2016*). This is accomplished by using all the previous squared values of the gradient. The update for each parameter $\theta_i$ in every iteration $t$ is expressed as follows:

$$\theta_{t,i} = \theta_{t-1,i} - \frac{\eta}{\sqrt{G_{t-1,ii} + \epsilon}} \cdot g_{t-1,i} \tag{12}$$

Here, $\epsilon$ is an expression that prevents division by zero, and the objective function's gradient with respect to the parameter $\theta_i$ at iteration $t-1$ is denoted with $g_{t-1}$:

$$g_{t-1,i} = \nabla_\theta J(\theta_i), \tag{13}$$

where $g_{t-1,ii}$ is a diagonal matrix where every diagonal element $i, i$ is the sum of gradients squares in respect to $\theta_i$ up to iteration $t-1$. Element-wise vector multiplication between $G_{t-1}$ and $g_{t-1}$ is represented with $\odot$, and vectorization update is done as follows:

$$\theta_t = \theta_{t-1} - \frac{\eta}{\sqrt{G_{t-1} + \epsilon}} \odot g_{t-1} \tag{14}$$

### AdaDelta

The primary goal of the AdaDelta method is to address the two main issues with AdaGrad: the constant decay of learning rates during training and the demand for manually selecting global learning rates. To achieve this, AdaDelta limits the past gradient window to a fixed size rather than adding up the sum of all squared gradients over time (*Zeiler, 2012*). The update of AdaDelta is as follows:

$$\theta_t = \theta_{t-1} - \frac{RMS[\Delta\theta]_{t-2}}{RMS[g]_{t-1}} g_{t-1} \tag{15}$$

where $RMS[g]_{t-1}$ represents the RMS error criterion of the gradient $[g]_{t-1}$:

$$RMS[g]_{t-1} = \sqrt{E[g^2]_{t-1} + \epsilon} \tag{16}$$

and the running average at the $t-1$th iteration is represented with $E[g^2]_{t-1}$, which is calculated using the previous average and the exist gradient:

$$E[g^2]_t = \gamma E[g^2]_{t-1} + (1-\gamma)g^2_{t-1} \tag{17}$$

where $\gamma$ denotes a fraction that is identical as the momentum term. Following this, $RMS[\Delta\theta]_t$ represents the error criterion of $\Delta\theta^2$, and the running average $E[\Delta\theta^2]_t$ of $\Delta\theta_t$ is acquired by:

$$E[\Delta\theta^2]_t = \gamma E[\Delta\theta^2]_{t-1} + (1-\gamma)\Delta\theta^2_{t-1} \tag{18}$$

### Adamax

Adamax is an extension of Adam. First, Adamax calculates the gradients at time step t with respect to the stochastic objective. Then, it calculates an exponentially weighted infinity norm along with a biased first-moment estimate to update model parameters (*Kingma & Ba, 2014*). Adamax's parameter update rule is as follows:

$$u_t = max(\beta_2 v_{t-1}, |g_t|) \tag{19}$$

$$\theta_{t+1} = \theta_t - \frac{\eta}{u_t}\hat{m}_t \tag{20}$$

### Nadam

Nadam uses the accelerated gradient of Nesterov to modify Adam's momentum component. Thus, the goal of Nadam is to increase the speed of convergence of the models (*Dozat, 2016*). Similar to the Adam algorithm, the first and second-moment variables are updated after computing the gradients. Following this, corrected moments are calculated so that the parameters can be updated. The parameter updates are expressed as:

$$\theta_{t+1} = \theta_t - \frac{\eta}{\sqrt{\hat{v}_t} + \epsilon}(\beta_1\hat{m}_t + \frac{(1-\beta_1)g_t}{1-\beta^{t_1}}) \tag{21}$$

## COMPARISON

In this study, we proposed a solution for surface classification tasks and evaluate the performance of the applied and proposed models. The evaluation process for the classification task in terms of training and testing is performed on the AMD Ryzen 9 5900X processor with 64 GB of RAM and Nvidia GeForce RTX 3080 Ti graphic card. Jupyter Notebook with Python 3.6 is used as an interface. Tensorflow and Keras libraries

are used for the implemented models. Matplotlib and Seaborn libraries are utilized for visualization.

In order to evaluate the model's accuracy, each model is trained and tested on a dataset containing 2,081 images, separated into training, validation, and test sets with 80-10–10%, respectively. Note that test images are selected randomly to predict the surface accurately. The efficiency and performance of the models are measured with four metrics: accuracy, precision, recall, and f1- score. Confusion matrices are demonstrated to analyze the model's predictions in detail. The performance evaluation metric equations are expressed below:

- The proportion of correctly identified samples to all samples is used to measure accuracy. This metric illustrates the classification's level of confidence.

$$Accuracy = \frac{TP + TN}{TP + FP + FN + TN} \tag{22}$$

- Precision is a metric that determines the ratio of true positives (TP) to the total of TP and false positives (FP).

$$Precision = \frac{TP}{TP + FP} \tag{23}$$

- Recall is the metric that demonstrates the ratio of the true positive to the total of the true positive and false negatives (FN).

$$Recall = \frac{TP}{TP + FN} \tag{24}$$

- The F-measure is a metric defined as the harmonic mean of precision and recall.

$$F1_{score} = \frac{2 \times Precision \times Recall}{Precision + Recall} \tag{25}$$

In Algorithm 1, first, the dataset, including images of carpet, tile, and wood, is loaded (line 1), and secondly, the images are normalized to reduce the computational cost (line 2). In the training, eight different models are trained (line 3), consisting of seven different state-of-the-art and one modified CNN-based model. Seven different optimizers (line 4) are used for each model. The models trained with different optimizers (line 10) are compared, and as a result, the model with the best performance (Proposed Model) is determined (line 11).

The train and validation accuracy/loss graphs obtained with the best-performing optimizers of each model are demonstrated between Figs. 6–13. When the figures are examined, it is seen that the train accuracy values of the models are above 96%. However, the values of the validation accuracy vary significantly according to the models. For instance, as shown in Figs. 6–8, although train accuracy is high in InceptionV3 and VGG19 models, validation accuracy could not exceed a specific limit. It implies that there is overfitting in both models. When analyzing the train-validation loss graphs of VGG19 and InceptionV3, it is observed that as the number of epochs increases during training, the train loss value of VGG19 decreases, whereas the InceptionV3 model increases.

Contrary to these models, it is seen that train and validation accuracy values in VGG16 and Xception models converge, and overfitting decreases compared to previous models, except for a few epochs when examining Figs. 7–10. This decrease in overfitting accuracy

**Algorithm 1** Surface classification in indoor environment

1: ImP=Load(Surface)
2: ImNorm=(ImP min(ImP))/(max(ImP)min(ImP))
3: *Models* ← { InceptionV3, VGG16, VGG19, Xception, ResNet50, InceptionResNetV2, MobileNetV2, Proposed}
4: *Optimizers* ← {SGD, Adam, RMSProp, Adadelta, Adamax, Adagrad, Nadam}
5: **for** i=1 to |*Models*| **do**
6:     **for** j=1 to |*Optimizers*| **do**
7:         TrainedModel[$i$][$j$]=Models[$i$](ImNorm,Optimizers[$j$])
8:     **end for**
9: **end for**
10: **for** k=1 to TrainedModel[$i$][$j$] **do**
11:     ProposedModel ← BestWeights(TrainedModel[$i$][$j$])
12: **end for**

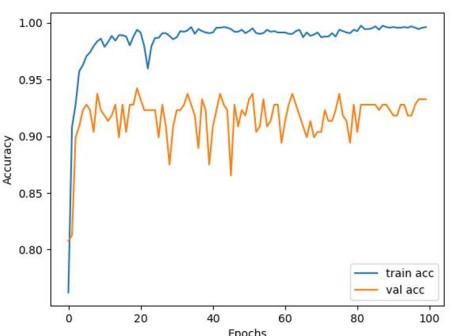
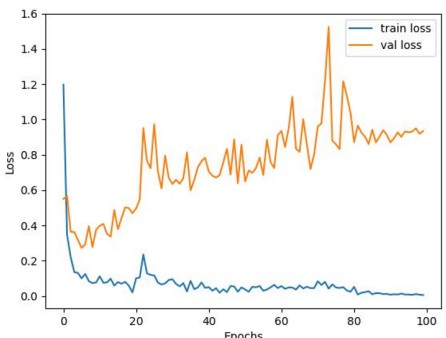

**Figure 6** **InceptionV3 model's training and validation accuracy-loss graphs with Adam optimizer.**

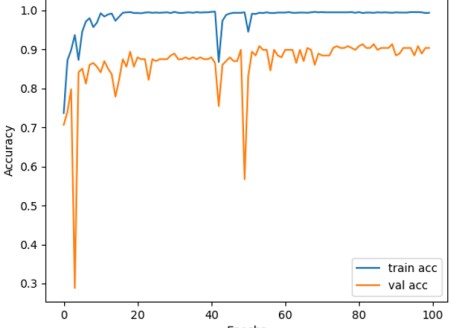
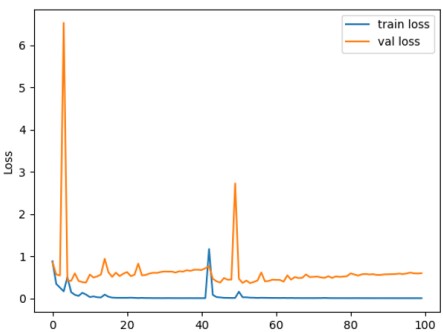

**Figure 7** **VGG16 model's training and validation accuracy-loss graphs with Nadam optimizer.**

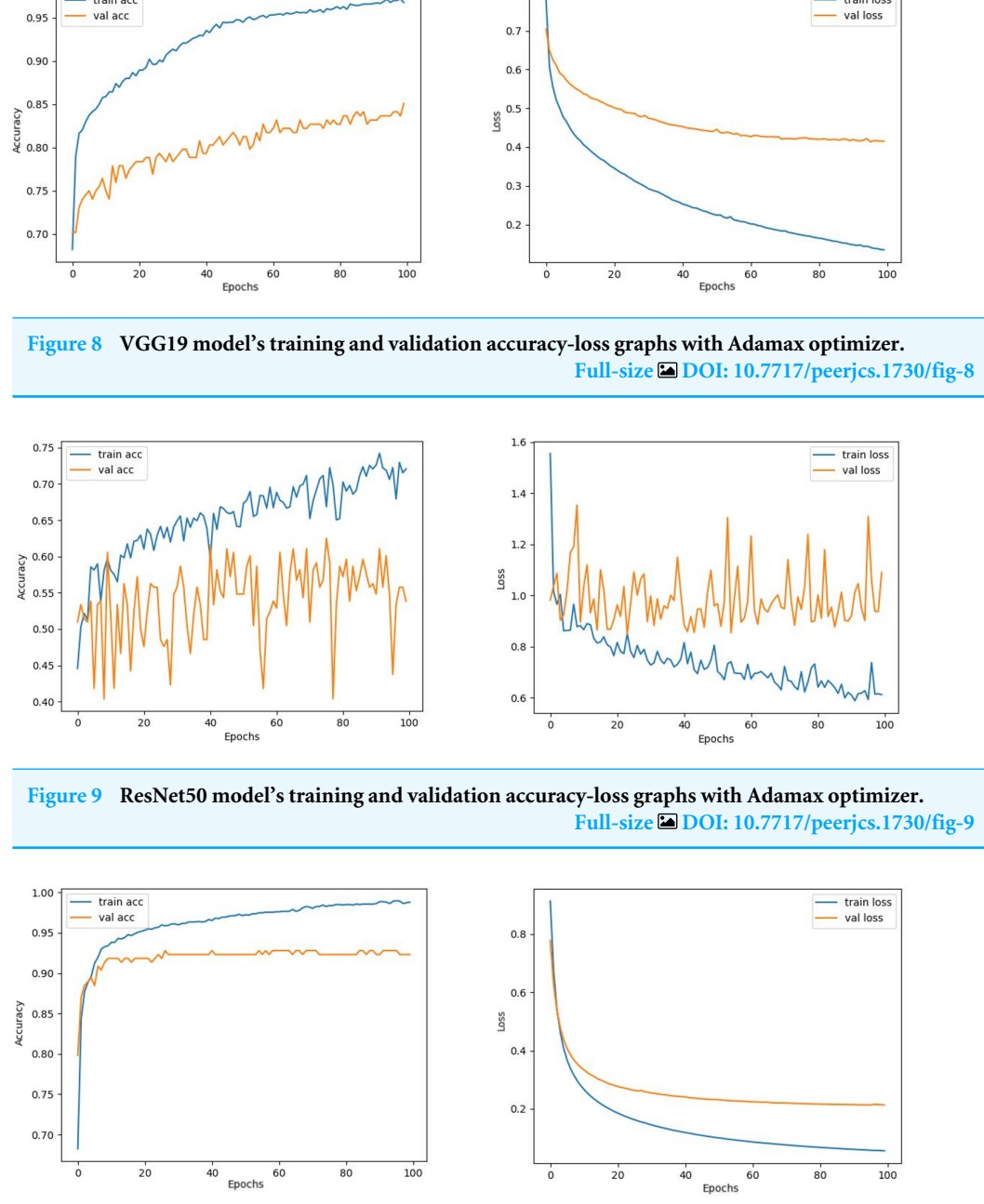

**Figure 8  VGG19 model's training and validation accuracy-loss graphs with Adamax optimizer.**

**Figure 9  ResNet50 model's training and validation accuracy-loss graphs with Adamax optimizer.**

**Figure 10  Xception model's training and validation accuracy-loss graphs with SGD optimizer.**

values is reflected in the graph as an increase in loss values. Although there is oscillation for the InceptionResNetV2 model in Fig. 11, the train and validation accuracy values are close. The train loss value is already low; the validation loss confirms this.

Comparing the MobileNetV2 and MobileNetV2-modified graphs in Figs. 12 and 13, it is evident that the training process of the proposed model is better than the original

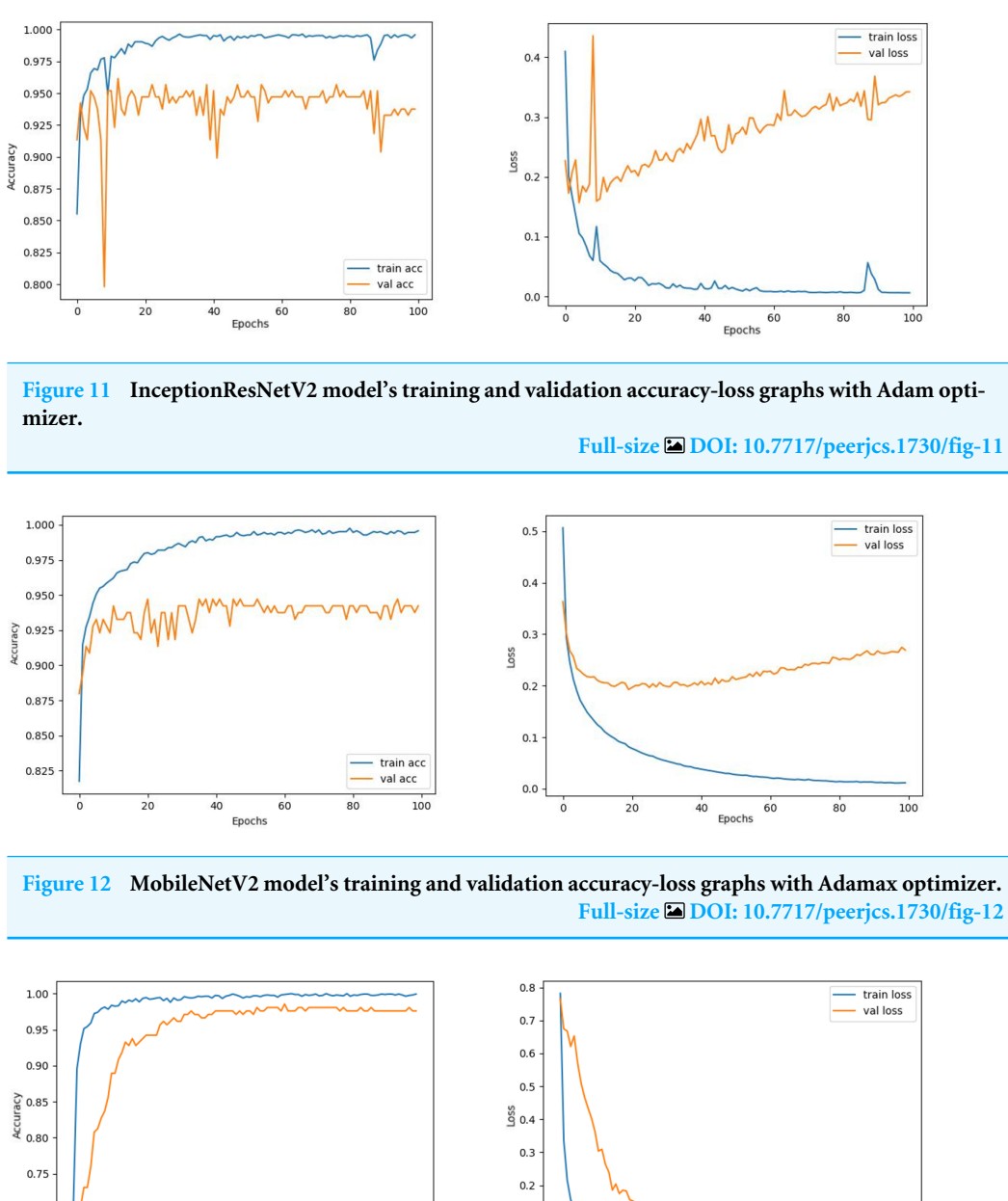

**Figure 11** InceptionResNetV2 model's training and validation accuracy-loss graphs with Adam optimizer.

**Figure 12** MobileNetV2 model's training and validation accuracy-loss graphs with Adamax optimizer.

**Figure 13** MobileNetV2-modified model's training and validation accuracy-loss graphs with Adamax optimizer.

MobileNetV2. Thus, this positively affects the proposed model's accuracy, precision, recall, and test accuracies.

The performance comparison analysis of the proposed model and applied state-of-the-art DL models for surface classification task is shown in Table 2. Experimental results

**Table 2  Performance results of applied state-of-the-art models and proposed model.**

| Model | Optimizer | Recall (%) | Precision (%) | F1-score (%) | Train Acc. (%) | Train loss (%) | Test Acc. (%) |
|---|---|---|---|---|---|---|---|
| | SGD | 95 | 95 | 95 | 99.40 | 3.8 | 95.19 |
| | Adam | 95 | 95 | 95 | 99.64 | 0.5 | 95.67 |
| | RMSprop | 96 | 95 | 95 | 99.58 | 5.04 | 95.67 |
| InceptionV3 | Adadelta | 92 | 93 | 92 | 95.74 | 14.45 | 92.79 |
| | Adamax | 95 | 95 | 95 | 99.52 | 2.56 | 95.67 |
| | Adagrad | 95 | 95 | 95 | 99.52 | 1.85 | 95.67 |
| | Nadam | 93 | 93 | 93 | 99.34 | 2.27 | 93.75 |
| | SGD | 86 | 87 | 86 | 88.77 | 35.42 | 87.02 |
| | Adam | 93 | 93 | 93 | 99.58 | 0.6 | 93.27 |
| | RMSprop | 92 | 92 | 92 | 99.46 | 0.7 | 91.83 |
| VGG16 | Adadelta | 84 | 85 | 85 | 85.29 | 44.57 | 85.58 |
| | Adamax | 93 | 93 | 93 | 99.52 | 0.64 | 93.27 |
| | Adagrad | 91 | 91 | 91 | 93.63 | 22.95 | 91.93 |
| | Nadam | 95 | 95 | 95 | 99.40 | 0.59 | 95.19 |
| | SGD | 87 | 87 | 87 | 84.98 | 42.69 | 87.98 |
| | Adam | 91 | 91 | 90 | 99.58 | 2.46 | 91.35 |
| | RMSprop | 91 | 91 | 91 | 99.76 | 1.42 | 91.83 |
| VGG19 | Adadelta | 52 | 56 | 49 | 59.76 | 93.38 | 57.69 |
| | Adamax | 94 | 94 | 94 | 96.76 | 13.47 | 94.23 |
| | Adagrad | 85 | 85 | 85 | 85.29 | 43.34 | 86.06 |
| | Nadam | 91 | 91 | 91 | 99.34 | 2.4 | 92.31 |
| | SGD | 47 | 38 | 35 | 56.05 | 95.21 | 41.29 |
| | Adam | 56 | 60 | 57 | 61.08 | 90.50 | 56.73 |
| | RMSprop | 54 | 55 | 46 | 66.85 | 70.62 | 53.85 |
| ResNet50 | Adadelta | 55 | 50 | 48 | 56.16 | 91.66 | 57.69 |
| | Adamax | 54 | 55 | 52 | 72.07 | 61.23 | 57.21 |
| | Adagrad | 55 | 53 | 50 | 61.26 | 80.94 | 57.21 |
| | Nadam | 54 | 59 | 54 | 65.53 | 70.84 | 58.17 |
| | SGD | 94 | 95 | 94 | 98.80 | 5.58 | 94.71 |
| | Adam | 93 | 93 | 93 | 99.58 | 0.55 | 93.75 |
| | RMSprop | 93 | 93 | 93 | 99.58 | 0.56 | 93.27 |
| Xception | Adadelta | 93 | 94 | 93 | 97.36 | 9.80 | 93.75 |
| | Adamax | 93 | 93 | 93 | 99.40 | 0.69 | 93.75 |
| | Adagrad | 93 | 93 | 93 | 99.46 | 2.25 | 93.27 |
| | Nadam | 33 | 14 | 20 | 41.92 | 109 | 41.35 |
| | SGD | 88 | 90 | 89 | 88.89 | 37.97 | 89.42 |
| | Adam | 95 | 95 | 95 | 99.58 | 0.61 | 95.19 |
| | RMSprop | 92 | 92 | 92 | 99.70 | 0.62 | 92.79 |
| InceptionResNetV2 | Adadelta | 86 | 88 | 86 | 85.83 | 67.83 | 87.50 |
| | Adamax | 94 | 94 | 94 | 99.52 | 1.26 | 94.71 |
| | Adagrad | 88 | 90 | 89 | 90.57 | 27.74 | 89.42 |
| | Nadam | 94 | 94 | 94 | 99.40 | 0.65 | 94.71 |

**Table 2** (*continued*)

| Model | Optimizer | Recall (%) | Precision (%) | F1-score (%) | Train Acc. (%) | Train loss (%) | Test Acc. (%) |
|---|---|---|---|---|---|---|---|
| | SGD | 91 | 92 | 91 | 89.01 | 36.55 | 91.83 |
| | Adam | 93 | 93 | 93 | 99.52 | 1.31 | 93.27 |
| | RMSprop | 95 | 95 | 95 | 99.46 | 1.62 | 95.19 |
| **MobileNetV2** | Adadelta | 79 | 82 | 80 | 79.52 | 69.71 | 81.25 |
| | Adamax | 95 | 95 | 95 | 99.58 | 1.15 | 95.67 |
| | Adagrad | 93 | 93 | 93 | 91.29 | 28.62 | 93.27 |
| | Nadam | 95 | 95 | 95 | 99.52 | 1.39 | 95.19 |
| | SGD | 95 | 96 | 95 | 87.45 | 38.80 | 95.67 |
| | Adam | 99 | 99 | 99 | 99.82 | 1.08 | 98.56 |
| | RMSprop | 99 | 99 | 99 | 100 | 0.002 | 99.04 |
| **MobileNetV2-modified** | Adadelta | 70 | 74 | 70 | 65.71 | 99.88 | 73.08 |
| | **Adamax** | **100** | **99** | **100** | **99.94** | **0.53** | **99.52** |
| | Adagrad | 97 | 97 | 97 | 93.09 | 21.95 | 97.12 |
| | Nadam | 99 | 99 | 99 | 100 | 0.01 | 98.56 |

**Notes.**
We emphasize the Modified model/Adamax values by making them bold, as they represent significant values, indicating the best performance in the table.

demonstrate that the modified model surpassed the state-of-the-art DL models in terms of precision, recall, F1-score, and test accuracy when trained with Adamax optimizer, by achieving 4% higher accuracy than each model. Afterward, the most effective ones are MobileNetV2, InceptionV3, Xception, and InceptionResNetV2 models, whereas VGG19 shows the lowest performance when considering the results obtained with different optimizers. Among the seven different optimizers, Adamax, Adam, SGD, and RMSProp perform the most effectively on each CNN model, while the others perform less effectively.

When Figs. 12 and 13 are compared, the effect of the modification can be seen quite clearly in terms of the accuracy and loss value. When the train-validation accuracy graph in Fig. 12 is examined, it is seen that the oscillation is lower, and there is a small difference in accuracy compared to the other models. As seen in Fig. 13, the training progress of the modified model has been completed consistently. In the train-validation accuracy and train-validation loss graphs, it is seen that the values converge between train-validation accuracy and loss. Thus, overfitting is prevented.

Consequently, the test accuracy is obtained as 99.52%, which is quite high compared to other models, and the loss value is close to 0 in Table 2. Therefore, this shows how well the modified model was trained. As seen in Fig. 14, MobileNetV2-modified mispredicts only one carpet image out of 208 test images in the confusion matrix, which reveals that the test accuracy is almost 100%.

The confusion matrices for each model are shown in Fig. 14, with each matrix representing the classification of images into 0 Carpet, 1 Tile, or 2 Wood. As shown in Fig. 14, 86 images of Carpet, 64 images of Tile, and 58 images of wood were tested. Each class was predicted with high accuracy, as indicated by the confusion matrix results. When examining the estimation of Carpet for each model, it is observed that the TP value is relatively high and the FP value is low, which is desirable. For instance, when the confusion matrix is examined for tile, the FN value is higher than other classes by reducing the recall

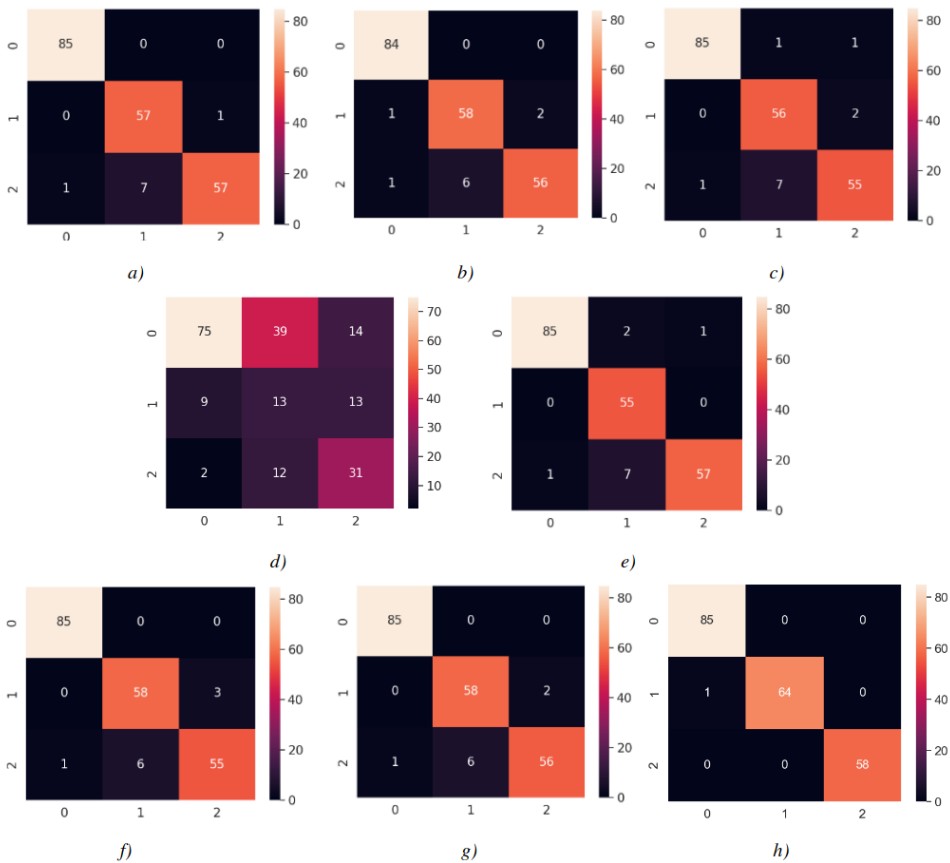

**Figure 14  Confusion matrices.** (A) InceptionV3 (Adam) (B) VGG16 (Nadam) (C) VGG19 (Adamax) (D) ResNet50 (Adamax) (E) Xception (SGD) (F) InceptionResnetV2 (Adam) (G) MobileNetV2 (Adamax) (H) MobileNetV2-modified (Adamax). Each model is associated with an optimizer performing best with the model.

value. It was determined as wood instead of being estimated as tile. This is because there are wood-like tiles and tile-like wood images in the section reserved for testing. In contrast, wood appears to be predicted more accurately. Considering the confusion matrices, the modified model has shown the highest prediction when compared to the applied DL models.

Additionally, in this study, the computational time of each model was also measured. Table 3 presents the training times and the weight sizes of both the modified model and the state-of-the-art DL models used in the surface classification task. Compared to the other models, the modified model effectively completed the training in less computing time. According to MobileNetV2, the training time increased partially as the depth of the proposed model increased. After removing blocks, adding dropouts, and freezing certain blocks, the trainable parameters have been reduced by approximately four times compared to the original MobileNetV2 and the weight of this model has been reduced to 11 MB. On the other hand, VGG19 and InceptionResNetV2 had the longest training time. It is

**Table 3  Comparison of training time and model weight size for each model.**

| Model | Training time (min) | Model weight size |
|---|---|---|
| InceptionV3 | 3:24 | 1.7 GB |
| VGG16 | 6:48 | 2.3 GB |
| VGG19 | 8:29 | 82 MB |
| ResNet50 | 7:13 | 1.4 GB |
| Xception | 6:47 | 927 MB |
| InceptionResNetV2 | 8:30 | 264 MB |
| MobileNetV2 | 1:43 | 42 MB |
| MobileNetV2-modified | 1:25 | 11 MB |

**Table 4  Comparison of the proposed model and existing studies.**

| Research | Sensor | Surface type | Algorithm/Technique | Accuracy (%) |
|---|---|---|---|---|
| *Kertész (2016)* | Ground Contact Force, Infrared, Accelerometer | Indoor | Random Forest (RF) | 94 |
| *Giguere & Dudek (2011)* | Tactile | Indoor/Outdoor | Artificial Neural Network (ANN) | 94.6 |
| *Lomio et al. (2019)* | IMU | Indoor | XGBoost+Neural Network+ResNet | 68.21 |
| *Singh et al. (2023)* | IMU | Indoor/Outdoor | Customized CNN Model | 88 |
| *Weiss, Tamimi & Zell (2008)* | Audio & Vision | Indoor/Outdoor | Support Vector Machine (SVM) | 87.04 |
| *Kurobe et al. (2021)* | Audio & Vision | Indoor/Outdoor | ResNet50 | 80 |
| *Guan et al. (2022)* | IMU & Vision | Indoor/Outdoor | EfficientNet-B0 + IMU Denoising Module | 98.37 |
| **Our study** | **Vision** | **Indoor** | **MobileNetV2-Modified** | **99.52** |

worth noting that the InceptionV3 model required a considerable amount of training time, whereas the training time for VGG16 and Xception models was on average.

It is worth noting that previous studies have mainly focused on surface classification tasks by using both IMU and vision sensors. Most of them utilize conventional machine learning algorithms or deep learning networks, which attain reasonable results. When Table 4 is examined, most of these studies were tested on indoor or outdoor surfaces by using more than one sensor. However, the accuracy remained within a certain limit except for the work (*Guan et al., 2022*). When they utilized only the vision sensor, the accuracy was around 90%, but when the IMU was added and the sensor fusion was supplied, it increased to around 98%. On the other hand, our study only used the vision sensor, and we facilitated its applicability in robotic applications by proposing a lightweight model called MobileNetV2-modifed. Consequently, with these advantages, the proposed model achieved an accuracy of 99.52%, which outperformed other studies.

## EXPERIMENTS

In this section, we first present the proposed approach (Algorithm 2) implemented on the mobile robot. Then, the characteristics of the utilized mobile robot are presented. Finally, we experimentally demonstrate the performance of the proposed model on the mobile robot under various indoor environments in surface classification tasks.

In Algorithm 2, first, the best weights obtained in the training are loaded to the proposed model (line 1). As the robot moves (line 2), it captures an image from the camera every five seconds (line 3). It tests the captured images using the best model weight obtained in training and generates probabilities for three surface types (line 4). Finally, the algorithm returns the surface type with the maximum probability (line 5).

---

**Algorithm 2** Real time surface classification in indoor environment

---
1: ProposedModel ← TrainedBestWeights
2: **while** Robot Moving **do**
3:     *image* ← CaptureImage in every 5 s
4:     *probs* ← ProposedModel.Predict(*image*)
5:     surface_type ← max(*probs*)
6: **end while**

---

## Experimental platform

The robot utilized for indoor surface classification is the Kobuki TurtleBot2, which is an open-source differential-drive robot developed by Yujin Robot. It has features such as long battery life, dependable odometry sensors, and power for external sensors and actuators, making it convenient for various indoor tasks. A laptop with an Intel i5 processor and 8 GB of RAM configuration is connected to the robot and placed on top of it.

For video streaming, we use a Logitech C922 webcam mounted on the laptop. With a 78-degree field of view, this webcam can capture 1080p HD videos at 30 frames per second (fps). The connectivity between the laptop and the robot is facilitated by Robot Operating System (ROS). ROS packages were utilized for the connectivity of the Logitech C922 camera to the Kobuki Turtlebot2 as well as for the keyboard control of the mobile robot. Note that the robot was controlled remotely *via* a keyboard from a computer that accessed the laptop. The experimental platform is shown in Fig. 15.

## Experimental results

To demonstrate the robustness of the proposed models in recognizing surface types, we conducted experiments on 11 different indoor environments. These environments include offices, rooms, and hallways. The surface types in each environment vary in terms of colors and patterns. We first transferred the weights of the proposed models (h5 file) to the laptop that was placed on top of the mobile robot. This made it suitable to perform surface classification on live streams during the robot is moving.

For the testing phase, 11 different scenarios were determined. Thus, we drove the mobile robot through passages where surfaces like wood-carpet-tile, carpet-tile, and carpet-wood were present. During the movement of the mobile robot, an image was captured from a Logitech camera every 5 s and classified as carpet, tile, or wood. In total, 267 images were captured and tested. The proposed model correctly classified the 265 surface images, while only two images were classified incorrectly. These images belong to scenario 8 in Table 5. Table 5 shows the number of correctly classified images for each scenario. In scenarios where one or two surface types did not exist it is presented with a dash (-).

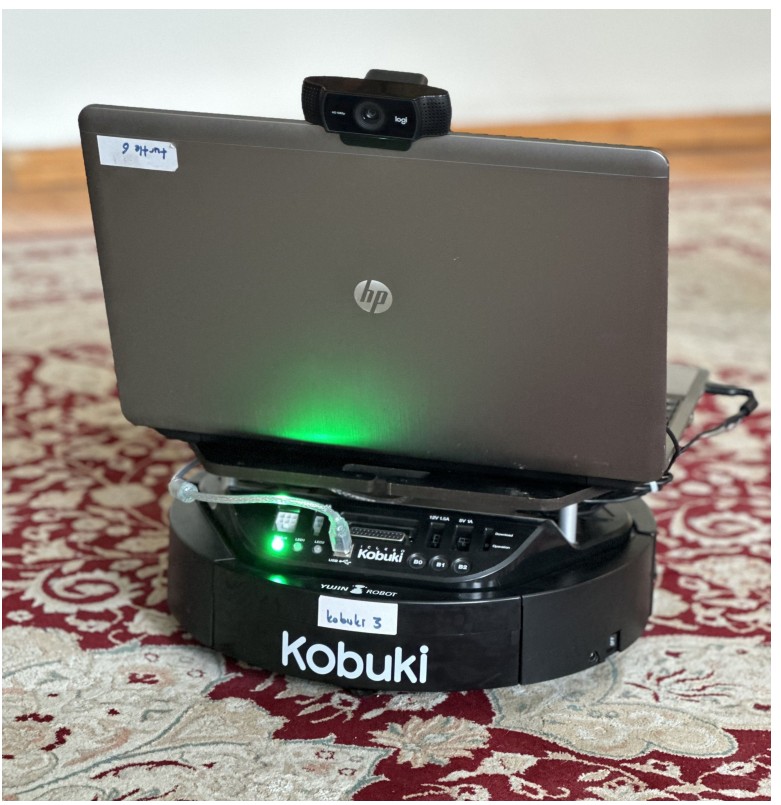

**Figure 15** Experimental platform.

**Table 5** Classification results of the proposed model implemented on a mobile robot.

| Scenario | Environment | Carpet | Tiles | Wood |
|----------|-------------|--------|-------|------|
| 1 | Hallway | 14 | 12 | – |
| 2 | | – | 15 | – |
| 3 | Office 1 | 11 | – | 17 |
| 4 | Office 2 | 9 | – | 13 |
| 5 | | 6 | 8 | 4 |
| 6 | Home 1 | 10 | 4 | 6 |
| 7 | | 14 | 10 | 7 |
| 8 | Home 2 | 10 | 8 | 10 |
| 9 | | 19 | 5 | 6 |
| 10 | Home 3 | 15 | – | 9 |
| 11 | | 10 | 8 | 7 |

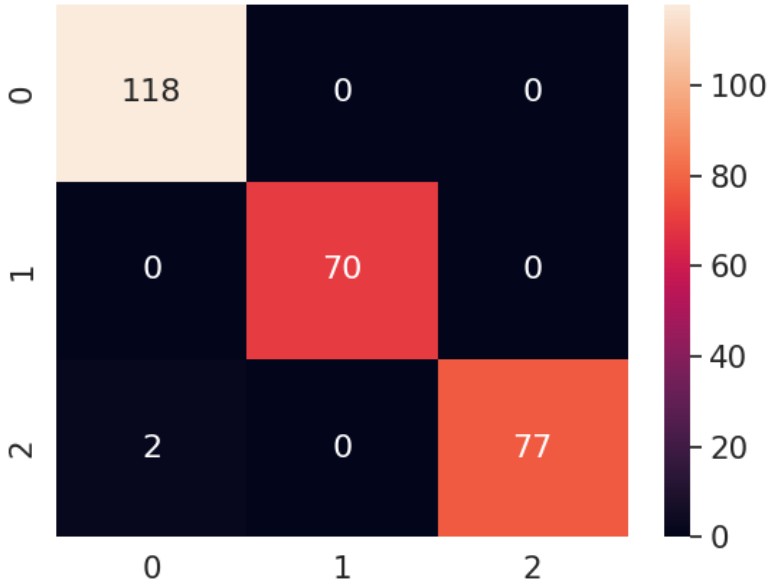

**Figure 16  Confusion matrix of proposed model for all the scenarios.**

The confusion matrix of the proposed model for all the scenarios is presented in Fig. 16. It is clear that the proposed model correctly predicts all of the images except for two images. In scenario 8, the proposed method classified the carpet surface type as wood. To investigate the source of the misclassification, we repeated the experiment for scenario 8 by moving the robot over the same area. However, this time, it classified all surface types correctly. After analyzing the images from the first experiment and the repeated experiment in scenario 8, we concluded that the misclassification was due to blurriness. Some of the correctly classified surface images and all misclassified images can be seen in Figs. 17 and 18, respectively.

In general, our model achieved an accuracy of 99.25% by correctly classifying 265 out of 267 surface images. Achieving high accuracy even in various indoor environments demonstrates the robustness of the proposed model.

The processing cost of the pre-trained models varies depending on the architecture and size of the model. We evaluated the processing cost of each model in terms of model loading time and processing time. When we refer to processing time, we imply the time required for predicting surface type. The evaluation was done on the laptop which we utilized on Kobuki robot and Raspberry Pi 3, a single board computer.

Table 6 presents the results obtained on an Intel laptop. As excepted, due to their architecture and model weight sizes, InceptionV3, VGG16, ResNet50, Xception and InceptionResNetV2 obtain a higher processing costs in terms of model loading and processing time. In comparison to the original MobileNetV2 model, the modified model requires less time for both model loading and processing time.

The results obtained using the Intel laptop reveal that, despite significant variations in model loading time for each architecture, the maximum processing time is 1.5 s. However,

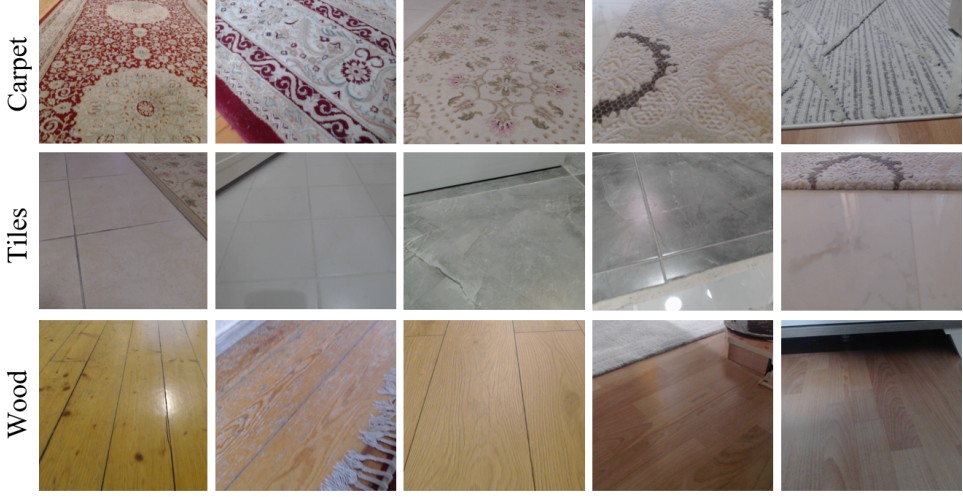

**Figure 17** Samples of correctly classified surface types.

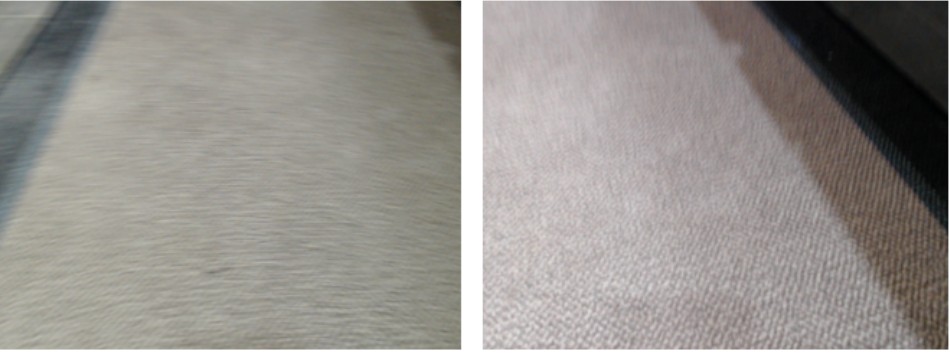

**Figure 18** Blurry images of carpet misclassified by the proposed model.

**Table 6** Processing cost of model weights on an Intel laptop.

| Model | Model loading time (sec.) | Processing time (sec.) |
|---|---|---|
| InceptionV3 | 8.44 | 0.37 |
| VGG16 | 11.76 | 1.28 |
| VGG19 | 0.61 | 1.52 |
| ResNet50 | 6.21 | 0.48 |
| Xception | 5.22 | 0.53 |
| InceptionResNetV2 | 7.73 | 0.70 |
| MobileNetV2 | 1.63 | 0.11 |
| MobileNetV2 Modified | 1.37 | 0.10 |

**Table 7  Processing cost of model weights on Raspberry Pi 3.**

| Model | Model loading time (sec.) | Processing time (sec.) |
|---|---|---|
| InceptionV3 | failed | – |
| VGG16 | failed | – |
| VGG19 | 15.87 | 7.91 |
| ResNet50 | failed | – |
| Xception | failed | – |
| InceptionResNetV2 | 168.09 | 114.02 |
| MobileNetV2 | 10.04 | 11.03 |
| MobileNetV2 Modified | 7.41 | 5.22 |

as seen in Table 7, there are significant differences in model loading time and processing time among the architectures when they were tested on Raspberry, which has a limited processing capacity. Although the evaluation process was performed multiple times for the InceptionV3, VGG16, ResNet50, and Xception architectures, Raspberry was unable to load them. InceptionResNetV2 and VGG19 are far behind both the original MobileNetV2 and MobileNetV2-modified in terms of model loading time and processing time. Compared to the original MobileNetV2, the processing and loading time has been significantly reduced in the MobileNetV2-modified model.

## CONCLUSION

Towards the improvement of indoor mobile robots for surface classification tasks, we developed a lightweight MobileNetV2-modified model. We then analyzed the performance of pre-trained state-of-the-art DL based models, including InceptionV3, VGG16, VGG19, ResNet50, Xception, InceptionResNetV2, and MobileNetV2. Our proposed model's total parameters are reduced approximately four times compared to the original MobileNetV2 model. This parameter reduction makes the model well-suited for use in computationally limited systems. Moreover, we generated a unique dataset with various types of carpets, tiles, and wood collected from different indoor environments. Finally, we applied several optimizers, namely SGD, RMSProp, Adam, Adadelta, Adamax, Adagrad, and Nadam, to determine the most efficient model. The experiments show that the proposed MobileNetV2-modified model outperforms all applied state-of-the-art DL models and existing approaches in the literature. Besides this, the robustness of the proposed model is demonstrated by its high accuracy performance in real-time indoor scenarios when examined on the mobile robot. Furthermore, it has been observed that the proposed model yields better results than other models in terms of model loading time and processing time on computers with limited computational capacity.

Future work includes enhancing the system by removing blurriness using image processing filters and computer vision techniques. In addition, different tasks will be assigned to heterogeneous robots based on the type of surface, allowing them to collaborate effectively. The robot that performs better on the relevant ground will complete the task. Therefore, this approach will facilitate task distribution based on surface conditions.

## ACKNOWLEDGEMENTS

The authors thank Erdal Alimovski for his helpful support during this work.

### Funding

This work was supported by Scientific Research Projects (BAP) through the Istanbul Sabahattin Zaim University (No. BAP-1000-88). The funders had no role in study design, data collection and analysis, decision to publish, or preparation of the manuscript.

### Grant Disclosures

The following grant information was disclosed by the authors:
Scientific Research Projects (BAP) through the Istanbul Sabahattin Zaim University: BAP-1000-88.

### Competing Interests

The authors declare there are no competing interests.

### Author Contributions

- Asiye Demirtaş conceived and designed the experiments, performed the experiments, analyzed the data, performed the computation work, prepared figures and/or tables, authored or reviewed drafts of the article, and approved the final draft.
- Gökhan Erdemir conceived and designed the experiments, analyzed the data, performed the computation work, prepared figures and/or tables, authored or reviewed drafts of the article, and approved the final draft.
- Haluk Bayram conceived and designed the experiments, analyzed the data, performed the computation work, prepared figures and/or tables, authored or reviewed drafts of the article, and approved the final draft.

### Data Availability

  The data is available at GitHub and Zenodo:
  - https://github.com/FieldRoboticsLab/Dataset-for-Indoor-Surface-Classification
  - Demirtas, A., Erdemir, G., & Bayram, H. (2023). Indoor Surface Classification for Mobile Robots [Data set]. Zenodo. https://doi.org/10.5281/zenodo.8415260

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
