# Peer review of "Indoor surface classification for mobile robots"

_PeerJ Computer Science, doi:10.7717/peerj-cs.1730_

## Round 0.1 · original submission · Minor Revisions

Please revise and improve according to the opinions of the two reviewers, and increase the rigor of the expression of the paper.

·

Basic reporting

The text is globally clear and very well written, easy to read.

Experimental design

I have some questions about the experimental design :

- choice of image size ? (line 151)
-> it is not clear for me, why you chose this size for the images (199x199). Do you mean that all the models that you test with your dataset have this input size ?
-> for me, this operation is called « resizing » and not « pre-processing »

- line 282 : « with 2D-convolutional layers » -> « with a 2D-convolutional layer » ?

- one of the main advantages of the modified MobileNetV2 is to reduce the processing cost.
However, you don't indicate the processing time (or, alternativey, Frame Per Second value), on the final system, which is a CPU. So it would be very interesting to have this information, as compared with the same result of the non-modified model. It would give useful information on the gain obtained in terms in processing cost.

- line 292 : the sentence is not clear for me : as in the modified model, there are more layers to train than in the original model, intuitively, learning would take more time in the modified model. But you say the contrary. Could you precise this ?

- in figure 5, it is not clear if block 13 is frozen or not.

- In the Experiments section, to my mind, the algorithm is not very relevant, firstly because it doesn't give new information compared to the previous sections, and secondly because learning and inference are not done on the same system, so that it is not relevant to put the learning process and the inference process in a single algorithm. So it could be removed without loss of clarity.

Validity of the findings

This work describes an interesting experimentation for a real-world robotic application.
But after reading the paper, one question remains : it is not very clear why the modified MobileNetV2 gets a better accuracy than the non-modified model. As I understand, it is due to the fact that you applied learning to more layers in the modified model. But that means that you could obtain better accuracy with the non-modified model if you do the learning from scratch ? It is difficult to understand why the pretrained MobileNetV2 would not be optimal for the task of surface classification. It would be useful do explicit more these aspects of the experimentations.

Cite this review as

Reviewer 2 ·

Basic reporting

The research significance and goal of this paper are clear, the research content and research method are put forward, the comparative experimental research is carried out fully, a more complete data set is established, and the proposed method has obtained better experimental results.

Experimental design

The experimental design is reasonable, compared with more mainstream deep learning methods, and proves the effectiveness of the proposed method.

Validity of the findings

The ability to recognize the surface type is crucial for both indoor and outdoor mobile robots. Knowing the surface type can help indoor mobile robots move more safely and adjust their movement accordingly. The proposed model can be used in robotic systems with limited computational capacity and embedded systems. Experimental results demonstrate that the proposed model outperforms all other applied methods and existing approaches in the literature by achieving 99.52% accuracy and an average score of 99.66% in precision, recall, and F1-score. Besides this, the proposed lightweight model was tested in real-time on a mobile robot in 11 scenarios consisting of various indoor environments such as offices, hallways, and homes, resulting in an accuracy of 99.25%.

Additional comments

How is the modified MobileNetV2 conducted. What are the improvements of the new model. It is suggested that the author give a more detailed and in-depth explanation on the innovative aspects of the network model to add persuasiveness to the creative contribution of this paper.

Cite this review as
Anonymous Reviewer (2024) Peer Review #2 of "Indoor surface classification for mobile robots (v0.1)". PeerJ Computer Science

---

## Round 0.2 · accepted · Accept

The author has answered all the questions raised by the reviewers in the last round and has made further modifications to the paper. I recommend accepting the paper.